# A pre-post evaluation study of a social media-based COVID-19 communication campaign to improve attitudes and behaviors toward COVID-19 vaccination in Tanzania

**Sooyoung Kim[1], Asad Lilani[2], Caesar Redemptus[3], Kate Campana[2], Yesim Tozan[1]***

**1** School of Global Public Health, New York University, New York, New York, United States of America, **2** The Access Challenge, New York, New York, United States of America, **3** Clouds Media Group, Dar Es Salaam, Tanzania

\* tozan@nyu.edu

**Data Availability Statement:** All data files are available from the OPEN ICPSR public repository (https://doi.org/10.3886/E198162V1).

## Abstract

In Tanzania, the One by One: Target COVID-19 campaign was launched nationally in July 2022 to address the prevalent vaccine hesitancy and lack of confidence in COVID-19 vaccines. The campaign mobilized social media influencers and viral content with the ultimate goal of increasing COVID-19 vaccine uptake in the country. The objective of this study was to empirically assess the impact of the campaign on three outcomes: vaccine confidence, vaccine hesitancy, and vaccination status. Using programmatic data collected through an online survey before and after the campaign, we conducted a difference-in-difference (DiD) analysis and performed a crude, adjusted, and propensity score-matched analysis for each study outcome. Lastly, to observe whether there was any differential impact of the campaign across age groups, we repeated the analyses on age-stratified subgroups. Data included 5,804 survey responses, with 3,442 and 2,362 responses collected before and after the campaign, respectively. Although there was only weak evidence of increased COVID-19 vaccine confidence in the campaign-exposed group compared to the control group across all age groups, we observed a differential impact among different age groups. While no significant change was observed among young adults aged 18–24 years, the campaign exposure led to a statistically significant increase in vaccine confidence (weighted/adjusted DiD coefficient = 0.76; 95% CI: 0.06, 1.5; p-value = 0.034) and vaccination uptake (weighted/adjusted DiD coefficient = 1.69.; 95% CI: 1.02, 2.81; p-value = 0.023) among young adults aged 25–34 years. Among adults aged 35 years and above, the campaign exposure led to a significant decrease in vaccine hesitancy (weighted/adjusted DiD coefficient = -15; 95% CI: -21, -8.3; p-value<0.001). The social media campaign successfully improved vaccine hesitancy, confidence, and uptake in the Tanzanian population, albeit to varying degrees across age groups. Our study provides valuable insights for the planning and evaluation of similar social media communication campaigns aiming to bolster vaccination efforts.

**Funding:** the author(s) received no specific funding for this work.

**Competing interests:** The authors have declared that no competing interests exist.

## Introduction

The COVID-19 pandemic's impact in Africa has been severe, with more than 8.9 million cases and about 174,000 lives lost as of May 2023 [1]. To date, COVID-19 vaccination coverage remains a challenge in low-income countries. Only 31% of the African population has been fully vaccinated, with great disparity across and within countries on the continent [2, 3], compared with 75% of people in high-income countries [3, 4]. The delayed roll-out of COVID-19 vaccines in African countries has left the population vulnerable to new surges in infections by COVID-19 variants throughout the pandemic. The situation has been further exacerbated by prevalent vaccine hesitancy and lack of confidence in COVID-19 vaccines, mainly fueled by exposure to inundating misinformation on COVID-19 and COVID-19 vaccines through online and other sources [5–7], against an already concerning backdrop of overall hesitancy toward new vaccines in Africa [8].

Tackling COVID-19 vaccine hesitancy and confidence is key to improving vaccine uptake in African countries and globally. Early in the COVID-19 pandemic, given the complexity of the public health crisis, national, regional, and global policymakers called for a "whole of society" approach, which aims to bring together all relevant stakeholders to develop integrated and coherent policies and actions to combat the pandemic [9]. In response to this call, the One-by-One (OBO): Target COVID-19, a pan-African communication and advocacy campaign, was launched in April 2020 by the African Union, the Africa CDC, the World Health Organization, and several other national, regional and global stakeholders, and is still ongoing. The principal aim of the campaign has been to support the Ministries of Health and the Africa CDC in their efforts to control the COVID-19 pandemic in African countries [10]. Initially, the campaign was focused on COVID-19 prevention messages and combating COVID-19-related misinformation. A continent-wide social media campaign called #AfricaCOVIDChampions was followed by country-specific campaigns in Democratic Republic of Congo (DRC), Zambia, and Uganda from April 2020 to early 2022 [11]. In early 2022, with COVID-19 vaccine accessibility no longer being a significant barrier in African countries, the thematic focus of the campaign shifted to address COVID-19 vaccine confidence and hesitancy with the ultimate goal of increasing COVID-19 vaccine uptake. Based on growing evidence supporting the effective utilization of social media platforms for public health communication [12, 13], the campaign mobilized influencers from all sectors of the society to use social media platforms and released viral content with key messages, such as "*COVID-19 vaccines are essential*", "*COVID-19 vaccines work*", "*COVID-19 vaccines are safe*", and "*Get informed, get vaccinated*", as well as educational content on how COVID-19 vaccines were developed, vaccine safety in pregnant women and people with comorbidities, and the importance of equitable access to vaccines.

A pilot campaign was launched in Tanzania in July 2022. The primary aim was to pilot the implementation of the social media-based campaign with the renewed focus prior to its implementation in other African countries. Tanzania served as an ideal setting for the pilot campaign for two reasons. A nationwide government-led COVID-19 vaccination campaign was launched in July 2021. A year later in June 2022, the percentage of the Tanzanian population fully vaccinated against COVID-19 was, however, only 7.3% [14]. Two of the identified barriers against COVID-19 vaccine roll-out were a high prevalence of misinformation and lack of accurate information on COVID-19 vaccines and the widespread apathy toward COVID-19 in the country [7, 15, 16]. On the other hand, as of 2021, roughly 50% of the Tanzanian population had access to the Internet, which was higher than the average internet penetration rate of 42% in African countries [17]. Encouragingly, a survey conducted in 2021 reported that, on a daily basis, about 13% of adult respondents used social media as a news source, indicative of the increasing penetration of social media platforms in the country [18].

During the pilot, routine monitoring and evaluation (M&E) data were collected alongside implementation before and after the campaign. In this study, using the routine M&E data, we evaluated the effects of the social media-based communication campaign on people's confidence and hesitancy toward COVID-19 vaccination in Tanzania. The evaluation findings will provide evidence and recommendations to implementers and decision-makers in promoting COVID-19 vaccine uptake through similar social-media based campaigns in settings where the prevalence of COVID-19 vaccine hesitancy is high.

## Methods

This study was a secondary analysis of the pre-and post-campaign data collected by the Access Challenge (TAC), the implementing agency of the campaign, for their internal monitoring and evaluation purposes. We aimed to empirically assess the impact of the One by One: Target COVID-19 social media pilot campaign in Tanzania. Specifically, we evaluated the extent to which the pilot campaign was effective in 1) decreasing COVID-19 vaccine hesitancy and 2) increasing confidence in COVID-19 vaccines, which, in turn, was expected to 3) improve the COVID-19 vaccine uptake in the target audience. We also examined whether the observed effects of the campaign differed across different age groups in the targeted population.

### Intervention: One by One: Target COVID-19 campaign

The One by One: Target COVID-19 campaign in Tanzania was launched nationwide in July 2022 and lasted until November 2022. The goal of the social media-based COVID-19 campaign was to support the Tanzanian Ministry of Health in reducing vaccine hesitancy and increasing vaccine confidence and hence promoting COVID-19 vaccine uptake in Tanzania. The main activity was the delivery of key messages on the importance of COVID-19 vaccination by trained high-profile and high-impact influencers on social media platforms, namely Twitter, Instagram, and Facebook. The key messages were developed prior to the launch of the campaign and were focused on debunking the myths about COVID-19 vaccines and on promoting the government COVID-19 vaccination hotline. As a result, the campaign was able to reach over 36 million people, resulting in about 998 million impressions through 4,948 relevant posts and 93,354 user engagements. In Dar es Salaam, the social media-based campaign was further supplemented by a number of community-based engagement activities targeting youth, community and faith leaders, and Bajaji (tuk tuk) drivers, and by special vaccination events. The toolkits used in the campaign are publicly available on the One by One: Target COVID-19 website (https://www.onebyone2030.org/targetcovid19-africa). In addition, S1 File presents the media and social media coverage of the campaign.

### Data collection: Measures and methods

We received de-identified data from the implementing agency for the purpose of this research, and an Institutional Review Board (IRB) review was deemed not required by New York University's IRB as presented in S2 File. The data used in this study were collected at two different time points, before and after the campaign, using an online survey platform and convenience sampling approach. Resultantly, the respondents to the pre- and post-campaign surveys were two distinct groups sampled from the same target population. The survey included questions that collected data on basic demographic and socioeconomic characteristics of respondents, including sex, age group, educational attainment, occupation, and region of residence in Tanzania. The survey also asked about respondents' self-reported vaccination status at the time of data collection, and individuals who had received two doses of the COVID-19 vaccine were considered 'fully vaccinated'. For those who responded that they were not fully vaccinated, an

additional question was asked to further ascertain if they had an appointment to complete the recommended full vaccination schedule. The pre- and post-surveys also measured respondents' COVID-19 vaccine confidence and hesitancy using validated scales that were successfully administered in previous studies [19, 20]. Specifically, we utilized the Oxford COVID-19 Vaccine Hesitancy Scale to assess vaccine hesitancy and the Oxford COVID-19 Vaccine Confidence and Complacency Scale to gauge vaccine confidence. Further information regarding the development and content of these scales can be found elsewhere [19, 20]. The vaccine hesitancy scale had 7 questions. Each question was rated between 0 and 5 (Cronbach's alpha = 0.83) with the total score ranging between 0 and 35. And higher scores indicating higher levels of hesitancy. The vaccine confidence scale included 5 questions. Each question was rated between 0 and 5 (Cronbach's alpha = 0.90) with the total score ranging between 0 and 25 and higher scores indicating higher levels of confidence. While all respondents were asked to respond to questions on vaccine confidence and uptake, vaccine hesitancy was measured only among those who self-reported not being fully vaccinated and having no plans to receive the full vaccination schedule at the time of survey. The survey questionnaire is included in Table S3-1 of S3 File.

Before the campaign, the baseline data (thereafter denoted as "pre-campaign") was collected during August 8–17, 2022. After the campaign, the end-line data (thereafter denoted as "post-campaign") was collected during November 16–30, 2022. During the data collection period, the survey was advertised nationwide on social media platforms by various influencers and promoted through radio advertisements. Informed consent was sought from all respondents using an electronic survey form at the time of the original data collection by the implementing agency, and only consented individuals participated in the survey. The online survey was conducted through a survey application specifically designed for the campaign using Code Rubik [21]. All participants were compensated for their time with mobile airtime. The final analytical sample included those who were 18 years or older and physically residing in Tanzania at the time of survey.

## Evaluation: Study design and statistical analysis

We conducted a pre-post outcome evaluation of the intervention using a difference-in-difference (DiD) approach to empirically investigate the campaign-attributed changes in COVID-19 vaccine confidence, hesitancy, and uptake. First, we merged the responses from the two survey rounds before and after the campaign, and created a dummy variable (*prepost*) that took the value of 1 for post-campaign responses, and 0 for pre-campaign responses. We created another dummy variable (*treatment*) that assigned respondents to a treatment or a control group based on their likely exposure to the campaign. At the time of conceiving this evaluation study, programmatic data collection at national level was already completed, and there was no clearly defined counterfactual group (i.e., a population not exposed to the campaign). Therefore, we used the information on respondents' main sources of COVID-19 information as a proxy to divide the respondents into a treatment and a control group. Specifically, since the campaign was rolled out only on social media platforms, we operationalized the survey question that probed respondents' main source of COVID-19-related information ("*What is your main source for acquiring COVID-19 information? (Check all that apply)*") to define the campaign exposure status. We assigned all those who self-reported social media as a main source to the treatment group (*treatment* = 1) and the rest to the control group (*treatment* = 0). We then performed descriptive analyses to explore the sample characteristics of the entire analytical sample and the sub-samples stratified by the *prepost* and *treatment* variables. We also

examined whether the sample characteristics differed between the groups using Student's t-test and chi-square test.

The main DiD analysis was conducted on three outcome variables—namely, vaccine confidence, vaccine hesitancy, and vaccination uptake. For each outcome variable, we conducted a crude and an adjusted analysis. In the crude analysis, we used the entire sample of non-missing observations on the outcome, and *prepost* and *treatment* variables. In the adjusted analysis, building upon the existing literature highlighting the relevance of socioeconomic factors as significant confounders of COVID-19 vaccine hesitancy, confidence, and uptake [22–25], we included in the model the potential confounders identified from the descriptive analyses by a significant *p-value* (i.e., age, gender, occupation, and educational attainment). In addition, we also matched the sample across the four groups, namely, pre/treatment, post/treatment, pre/control, post/control, using the propensity score matching method, and conducted a matched analysis using the same set of confounders included in the adjusted analysis [26]. Lastly, we conducted age-stratified analyses to observe whether the effects of the campaign differed by age group. For these analyses, we divided the survey respondents into three age groups: ages 18–24 years, 25–34, and 35 and above. All analyses were conducted using R (version 4.2.2). The Strengthening the Reporting of Observational Studies in Epidemiology (STROBE) checklist is provided in Table S4 of S4 File.

## Results

### Descriptive statistics

Table 1 presents the sample characteristics stratified by the *prepost* and *treatment* variables. A total of 5,804 responses were collected over two survey rounds and included in the analysis. Of the 5,804 responses, 3,442 were collected prior to the campaign, and 2,362 were collected after the campaign. Based on respondents' self-reported main source of COVID-19 information, 4,846 respondents (83.5%) were categorized as likely to have been exposed to the campaign. The respondents were predominantly male (85.7%) and younger than 35 years old (90.7%). More than 40% of respondents were residing in Dar es Salaam region, followed by Dodoma (7.0%), Arusha (6.5%), and Mwanza (5.4%) region. The majority of respondents (64.0%) were employed (full-time, part-time, or self-employed) while 20.9% were unemployed, and 14.0% were students. Similarly, 64.9% of respondents had educational attainment at or above to the diploma level, which is equivalent to the Advanced Certificate of Secondary Education (ACSE) corresponding to grades 13–14 [27]. When stratified by campaign exposure status and survey round, the sample characteristics across the four groups differed significantly in terms of sex (*p-value* = 0.005), age distribution (*p-value* <0.001), occupation (*p-value* = 0.006), and educational attainment (*p-value*<0.001). Tables S5-1~3 in S5 File present the sample's descriptive statistics based on the pre/post stratification and treatment/control stratification at baseline and endline surveys separately.

The last three rows in Table 1 and Fig 1 present the crude average and the trend of the outcome variables before and after the campaign stratified by campaign exposure status. Additionally, the frequencies and percentages for individuals who chose each category in the Vaccine Hesitancy and Vaccine Confidence scales are summarized in the Tables S5-5 and S5-6 of S5 File. All three outcomes of interest, namely, COVID-19 vaccine confidence, hesitancy, and proportion of fully vaccinated individuals, appeared to have increased over time. The results also showed that, on average, the treatment group had a lower level of COVID-19 vaccine confidence than the control group prior to the campaign. However, this crude difference was narrowed following the campaign (Fig 1A). In contrast, the difference in COVID-19 vaccine hesitancy between the two groups increased following the campaign, with a lower increase

**Table 1. Overall sample characteristics of respondents by treatment status before and after the social-media based COVID-19 campaign in Tanzania.**

| | Pre-campaign (N = 3,442) | | Post-campaign (N = 2,362) | | Overall (N = 5,804) | p-value |
|---|---|---|---|---|---|---|
| | Control (N = 569) | Treatment (N = 2,873) | Control (N = 389) | Treatment (N = 1,973) | | |
| **Sex** | | | | | | 0.005 |
| Female | 76 (13.4%) | 369 (12.8%) | 60 (15.4%) | 323 (16.4%) | 828 (14.3%) | |
| Male | 493 (86.6%) | 2504 (87.2%) | 329 (84.6%) | 1650 (83.6%) | 4976 (85.7%) | |
| **Age group** | | | | | | <0.001 |
| 18–24 | 151 (26.5%) | 773 (26.9%) | 105 (27.0%) | 656 (33.2%) | 1685 (29.0%) | |
| 25–34 | 330 (58.0%) | 1795 (62.5%) | 253 (65.0%) | 1203 (61.0%) | 3581 (61.7%) | |
| 35–44 | 67 (11.8%) | 255 (8.9%) | 21 (5.4%) | 101 (5.1%) | 444 (7.6%) | |
| 45–54 | 17 (3.0%) | 41 (1.4%) | 5 (1.3%) | 8 (0.4%) | 71 (1.2%) | |
| 55–64 | 3 (0.5%) | 6 (0.2%) | 3 (0.8%) | 4 (0.2%) | 16 (0.3%) | |
| 65 and over | 1 (0.2%) | 3 (0.1%) | 2 (0.5%) | 1 (0.1%) | 7 (0.1%) | |
| **Region of residence** | | | | | | 0.791 |
| Dar es salaam region | 231 (40.6%) | 1262 (43.9%) | 159 (40.9%) | 853 (43.2%) | 2505 (43.2%) | |
| Dodoma region | 45 (7.9%) | 182 (6.3%) | 32 (8.2%) | 149 (7.6%) | 408 (7.0%) | |
| Arusha region | 40 (7.0%) | 188 (6.5%) | 31 (8.0%) | 120 (6.1%) | 379 (6.5%) | |
| Mwanza region | 31 (5.4%) | 152 (5.3%) | 21 (5.4%) | 112 (5.7%) | 316 (5.4%) | |
| Mbeya region | 23 (4.0%) | 140 (4.9%) | 16 (4.1%) | 78 (4.0%) | 257 (4.4%) | |
| Kilimanjaro region | 15 (2.6%) | 98 (3.4%) | 14 (3.6%) | 64 (3.2%) | 191 (3.3%) | |
| Other regions | 184 (32.3%) | 851 (29.6%) | 116 (29.8%) | 597 (30.3%) | 1748 (30.1%) | |
| **Occupation** | | | | | | 0.006 |
| Employed full-time | 101 (17.8%) | 409 (14.2%) | 50 (12.9%) | 231 (11.7%) | 791 (13.6%) | |
| Employed part-time | 106 (18.6%) | 526 (18.3%) | 96 (24.7%) | 388 (19.7%) | 1,116 (19.2%) | |
| Housekeeper | 2 (0.4%) | 10 (0.3%) | 1 (0.3%) | 7 (0.4%) | 20 (0.3%) | |
| Retired | 2 (0.4%) | 3 (0.1%) | 2 (0.5%) | 1 (0.1%) | 8 (0.1%) | |
| Self-employed | 160 (28.1%) | 925 (32.2%) | 116 (29.8%) | 644 (32.6%) | 1845 (31.8%) | |
| Student | 86 (15.1%) | 403 (14.0%) | 41 (10.5%) | 283 (14.3%) | 813 (14.0%) | |
| Unemployed | 112 (19.7%) | 597 (20.8%) | 83 (21.3%) | 419 (21.2%) | 1,211 (20.9%) | |
| **Educational attainment** | | | | | | <0.001 |
| No qualification | 3 (0.5%) | 7 (0.2%) | 3 (0.8%) | 5 (0.3%) | 18 (0.3%) | |
| Primary school | 20 (3.5%) | 103 (3.6%) | 19 (4.9%) | 88 (4.5%) | 230 (4.0%) | |
| Secondary school | 113 (19.9%) | 579 (20.2%) | 114 (29.3%) | 572 (29.0%) | 1378 (23.7%) | |
| A-level secondary school | 44 (7.7%) | 217 (7.6%) | 23 (5.9%) | 130 (6.6%) | 414 (7.1%) | |
| Diploma* | 151 (26.5%) | 729 (25.4%) | 109 (28.0%) | 531 (26.9%) | 1,520 (26.2%) | |
| University degree or higher | 238 (41.8%) | 1238 (43.1%) | 121 (31.1%) | 647 (32.8%) | 2,244 (38.7%) | |
| **Vaccine confidence score (Range 0–25)** | | | | | | 0.069 |
| Mean (SD) | 22.8 (3.31) | 22.3 (3.99) | 23.0 (3.68) | 23.0 (3.25) | 22.7 (3.67) | |
| **Vaccine hesitancy score (Range 0–35)** | | | | | | 0.681 |
| Mean (SD) | 21.7 (7.31) | 21.6 (6.80) | 22.9 (7.59) | 22.2 (7.52) | 21.9 (7.12) | |
| **Vaccination status** | | | | | | <0.001 |
| Not fully vaccinated | 121 (21.3%) | 827 (28.8%) | 65 (16.7%) | 377 (19.1%) | 1390 (23.9%) | |
| Fully vaccinated + partially vaccinated with confirmed appointment(s) | 252 (44.3%) | 1,033 (36.0%) | 194 (49.9%) | 838 (42.5%) | 2317 (39.9%) | |
| Missing | 196 (34.4%) | 1,013 (35.3%) | 130 (33.4%) | 758 (38.4%) | 2097 (36.1%) | |

* Diploma in the Tanzanian school system is equivalent to the Advanced Certificate of Secondary Education (ACSE) corresponding to grades 13–14 [27]

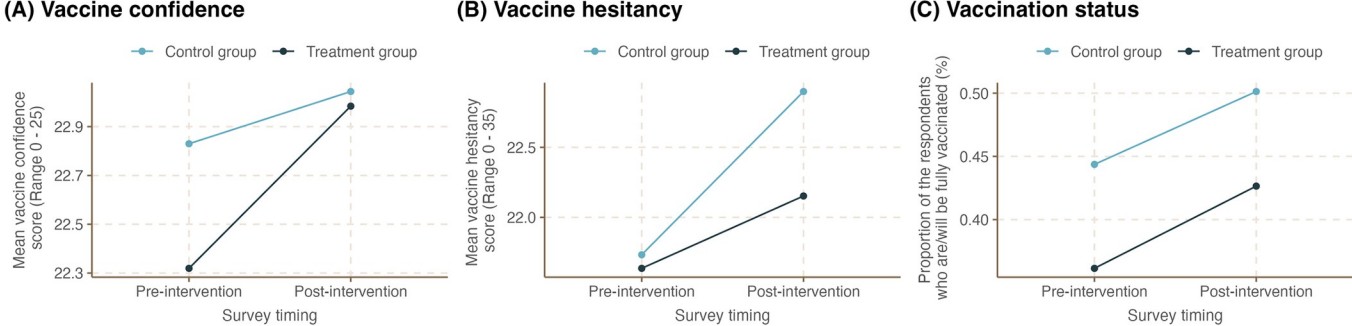

**Fig 1. COVID-19 vaccine confidence, hesitancy, and vaccination status among the treatment and control group respondents before and after the social-media based COVID-19 campaign in Tanzania.**

among individuals in the treatment group (Fig 1B). The difference in vaccination status across the two groups seemed to, however, have remained the same throughout the study period (Fig 1C). The observed pre-post changes in all three outcome variables stratified by age group are presented in Fig S5-1 of S5 File.

## Evaluation findings

We evaluated the effects of the social media-based campaign on COVID-19 vaccine hesitancy, vaccine confidence, and vaccine update across all ages and also by age group. Tables S5-7~9 in S5 File and Fig 2 present the results of the difference-in-difference (DiD) analyses that quantified the campaign-attributed changes in the three outcome variables across all age groups. We observed weak evidence of increased COVID-19 vaccine confidence by about additional 0.5 points in the treatment group over the control group following the campaign (Table S5-8 in S1 File, Adjusted DiD coefficient = 0.47; 95% CI: -0.07, 1.0; *p-value* = 0.091, Weighted/adjusted DiD coefficient = 0.52; 95% CI: -0.08, 1.1; *p-value* = 0.087). Among individuals who did not, or had no plan to, receive a full dose of COVID-19 vaccination, no significant campaign-attributed change in COVID-19 vaccine hesitancy was observed following the campaign (Table S5-7 in S1 File, Adjusted DiD coefficient = -0.22; 95% CI: -2.7, 2.3; *p-value* = 0.864, Weighted/adjusted DiD coefficient = -0.41; 95% CI: -3.1, 2.3; *p-value* = 0.766). No campaign-attributed significant difference in the proportion of fully vaccinated people was also observed across the two groups (Table S5-9 in S5 File, Adjusted DiD coefficient = 1.25; 95% CI: 0.84, 1.85; p-value = 0.267, Weighted/adjusted DiD coefficient = 1.06; 95% CI: 0.98, 1.15, p-value = 0.156).

Figs 3–5 and Tables S5-10~18 in S5 File present the results of age-stratified DiD analyses that quantified the campaign-attributed changes in the three outcome variables by age group. In this age-stratified analysis, we observed a differential impact of the campaign on the study outcomes. While no significant campaign-attributable change was observed for vaccine confidence, hesitancy, and uptake among young adults aged 18–24 years (Fig 3, Table S4 in S4 File), campaign exposure was associated with a statistically significant increase in vaccine confidence and vaccination uptake among those aged 25–34 years (Fig 4, Table S4 in S4 File). Specifically, vaccine confidence in this age group increased by 0.76 points in the treatment group over the control group following the campaign (Table S4 in S4 File, Weighted/adjusted DiD coefficient = 0.76; 95% CI: 0.06, 1.5; *p-value* = 0.034). Similarly, vaccination uptake in this age group increased by 1.6% in the treatment group over the control group following the campaign (Table S4 in S4 File, Adjusted DiD coefficient = 1.64; 95% CI: 1.00, 2.68; *p-value* = 0.049, Weighted/adjusted DiD coefficient = 1.69.; 95% CI: 1.02, 2.81; *p-value* = 0.023). Further, we

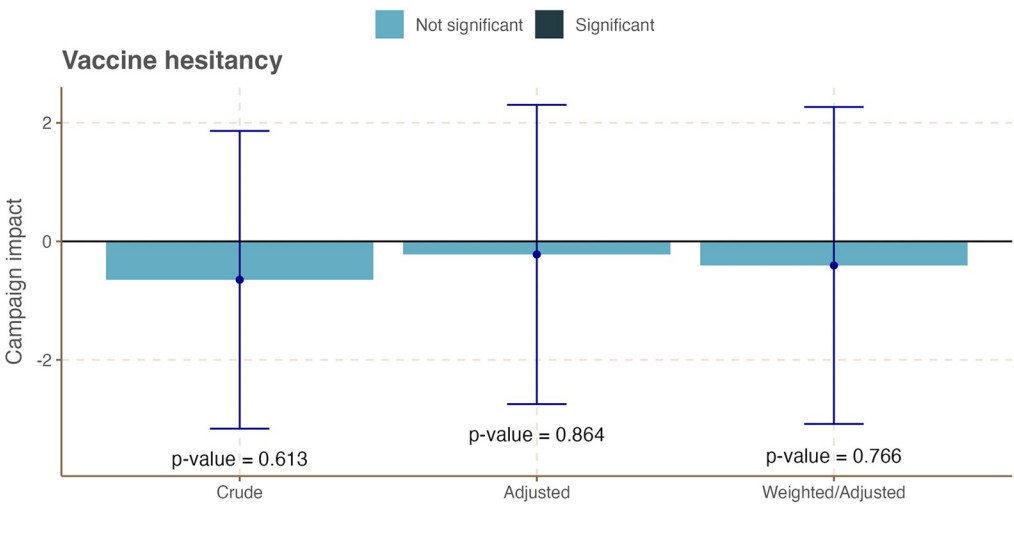

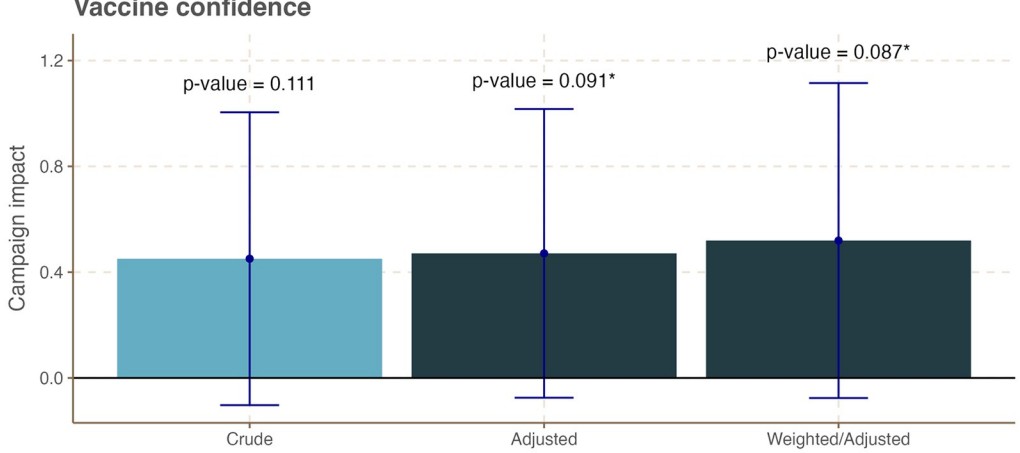

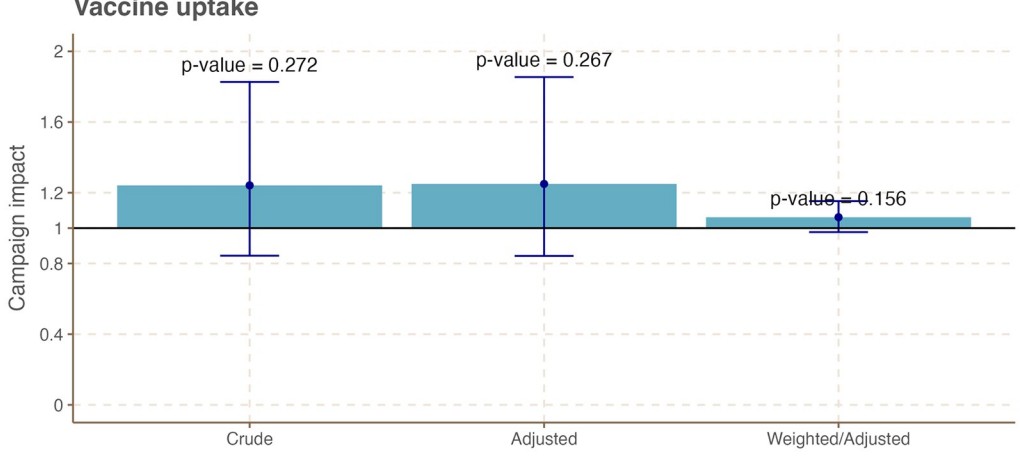

**Fig 2. Summary plot of campaign-attributable changes on COVID-19 vaccine confidence, hesitancy, and uptake across all age groups in Tanzania.**

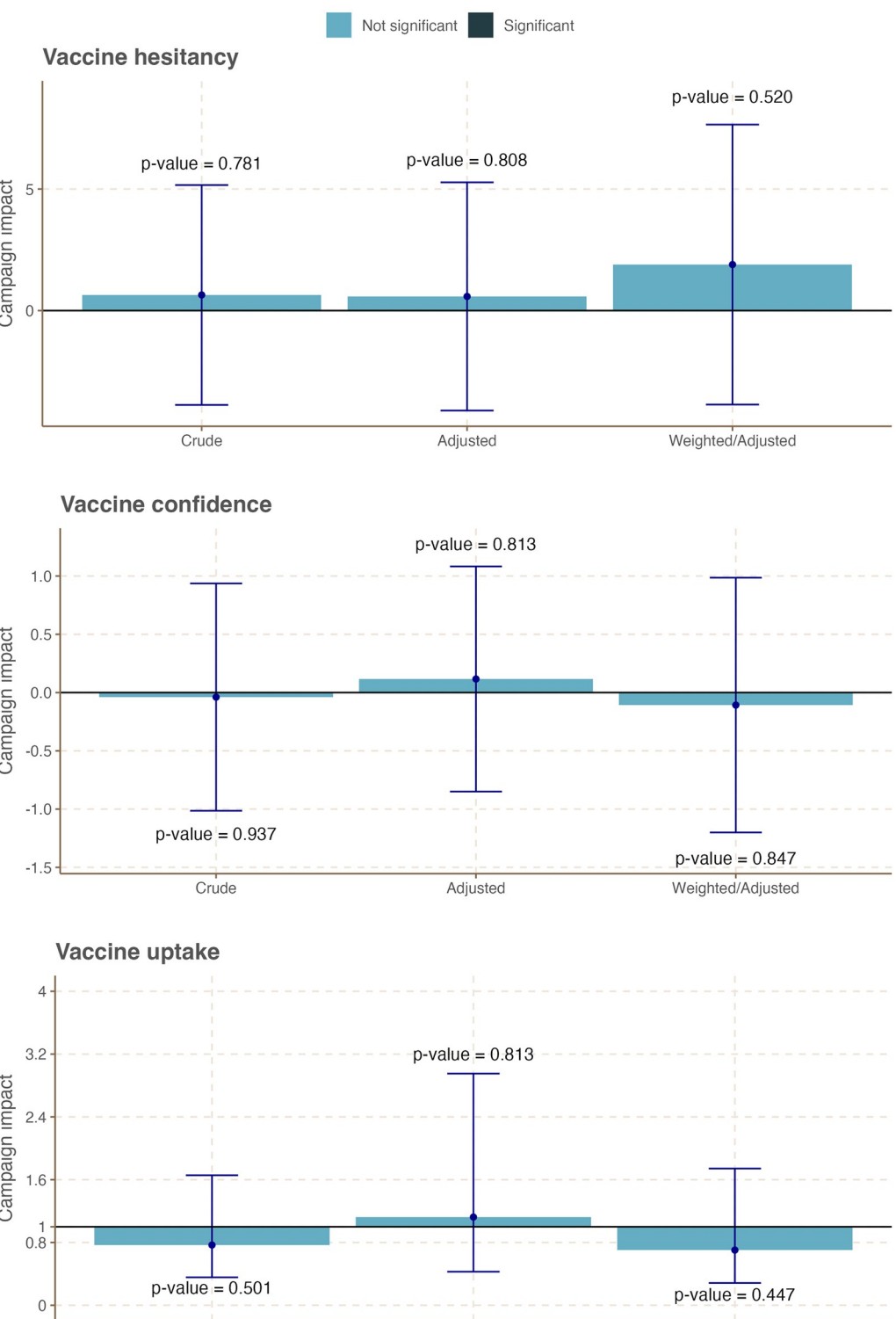

**Fig 3. Summary plot of campaign-attributable changes on COVID vaccine confidence, hesitancy, and vaccination uptake among respondents aged 18–24 years in Tanzania.**

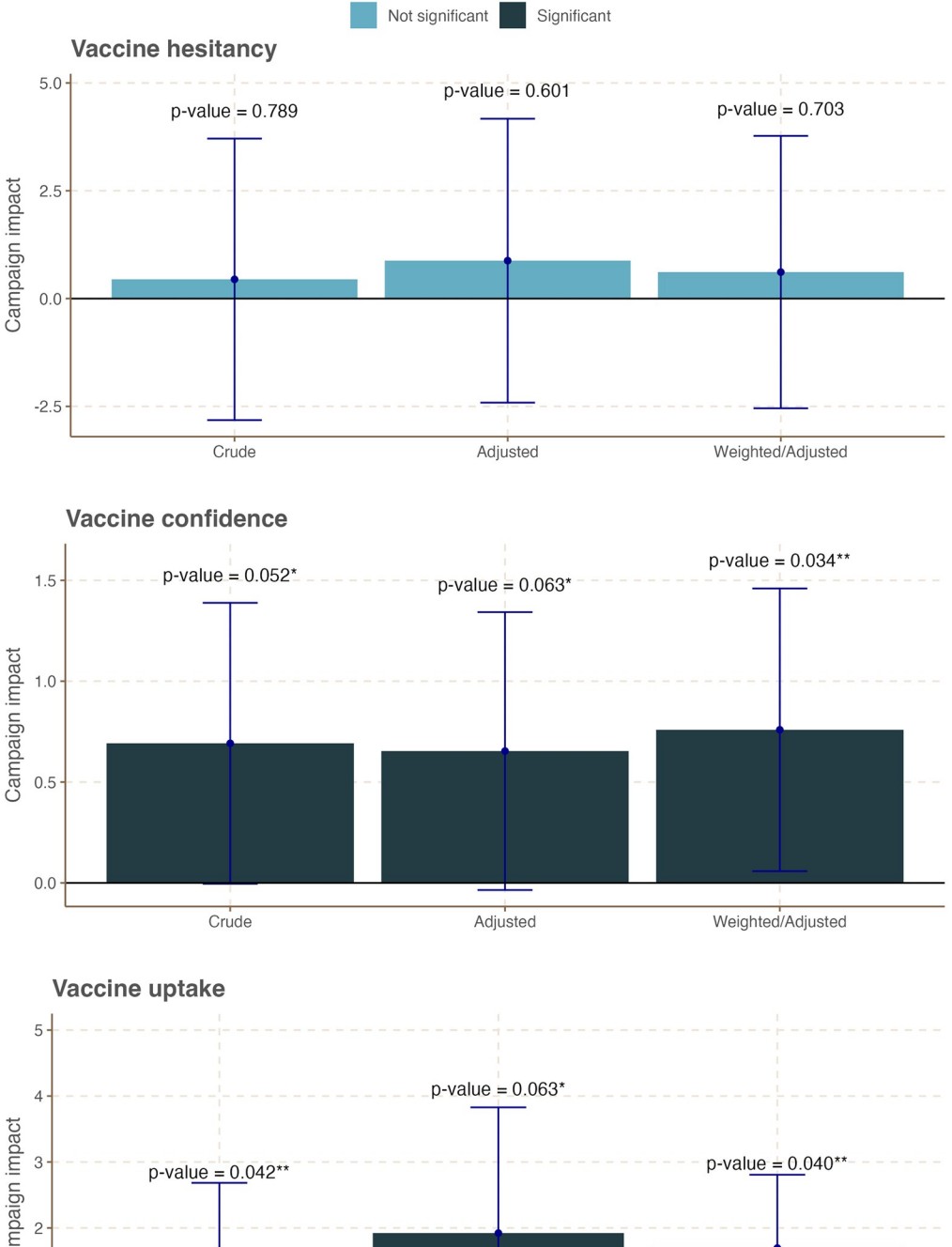

**Fig 4. Summary plot of campaign-attributable changes on COVID-19 vaccine confidence, hesitancy, and vaccination uptake among respondents aged 25–34 years in Tanzania.**

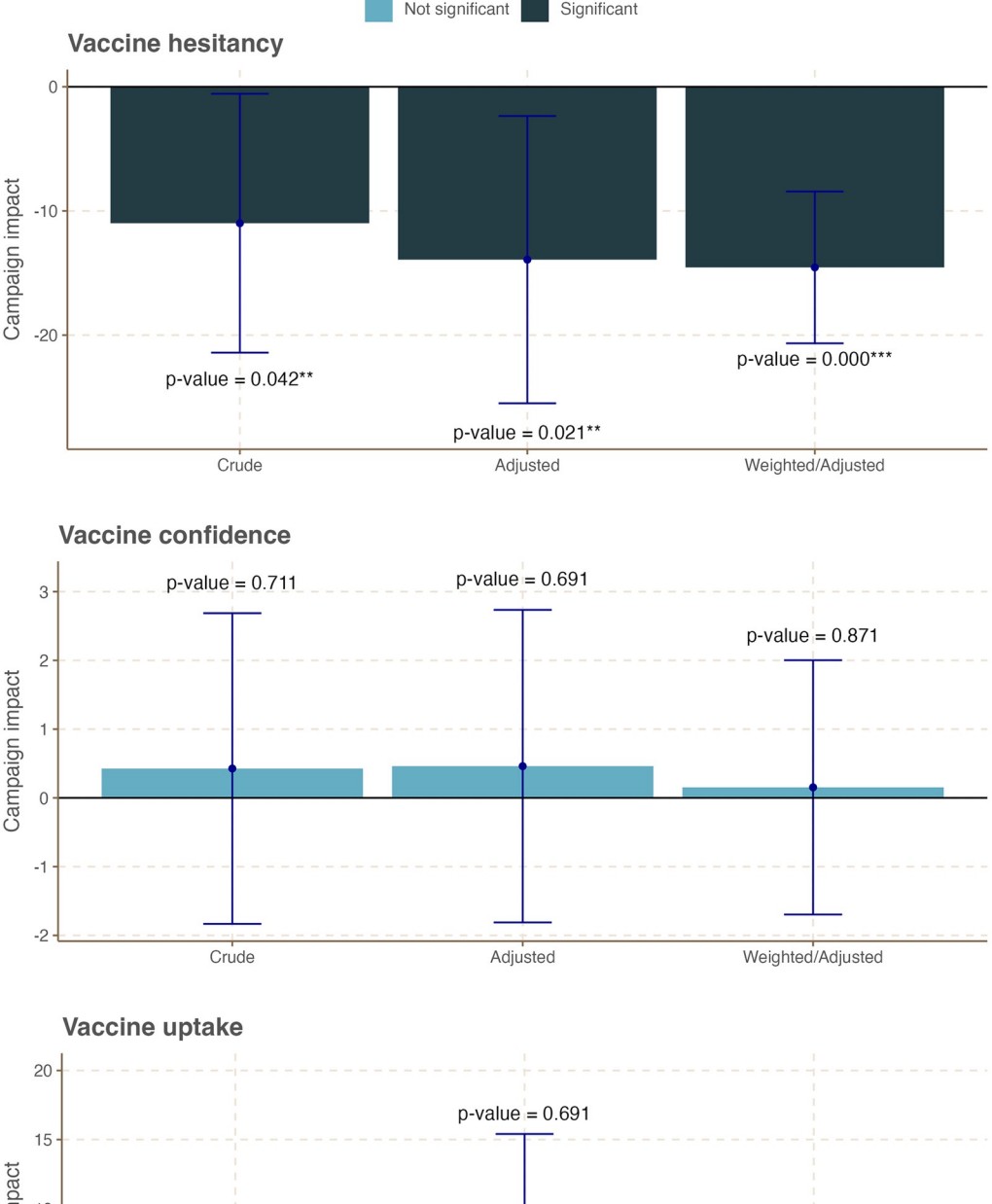

**Fig 5. Summary plot of campaign-attributable changes on COVID-19 vaccine confidence, hesitancy, and vaccination uptake among respondents aged 35 years and over in Tanzania.**

observed a significant decrease in vaccine hesitancy among adults aged 35 years and over across the two groups following the campaign (Fig 5, Table S4 in S4 File, Adjusted DiD coefficient = -14; 95% CI: -25, -2.4; *p-value* = 0.021, Weighted/adjusted DiD coefficient = -15.; 95% CI: -21, -8.3; *p-value* = <0.001). In summary, the campaign appeared to have significantly improved the target population's COVID-19 vaccine hesitancy, vaccine confidence, and vaccination uptake, albeit to different degrees across different age groups.

## Discussion

In Tanzania and other African countries, COVID-19 vaccine rates remained lower compared to other low- and middle-income regions in the beginning of the vaccine roll-out. This was attributed, in part, to vaccine accessibility and, in part, to vaccine hesitancy due to widespread misinformation about vaccine origin, safety and efficacy [28]. In this study, we attempted to empirically evaluate the effects of a social media-based communication campaign on COVID-19 vaccine hesitancy, confidence, and uptake in Tanzania, using data collected through validated vaccine confidence and hesitancy scales and employing a quasi-experimental analytical framework. Keeping the limitations of the evaluation design in mind, which are discussed at length below, the campaign did not appear to have an impact on COVID-19 vaccine hesitancy or vaccine uptake across all age groups. However, we observed weak and inconclusive evidence (*p-value*<0.10) suggestive of an increase in vaccine confidence in the target population. Encouragingly, when the campaign's effects were analyzed by age group, results showed differential and significant effects, and all the effects were in a positive direction. Specifically, while the campaign had no significant effect on vaccine confidence, vaccine hesitancy, and vaccine uptake among young adults aged 18–24 years, campaign exposure was associated with a significant increase in vaccine confidence among those aged 25–34 years. Furthermore, the campaign was associated with a significant decrease in vaccine hesitancy among older respondents aged 35 years and above.

The evaluation shed light on the level of vaccine hesitancy and confidence in the target population. Our results showed an average hesitancy score of 21.9 (SD = 7.12) among survey respondents, which was significantly higher than that of other studies conducted in different countries using the same COVID-19 vaccine hesitancy scale. For instance, a study conducted in Malaysia in May-June 2021 reported an average hesitancy score of 11.3 (SD = 4.39), while studies in Turkey, Pakistan (among male population only), and Australia (among unvaccinated individuals with underlying medical conditions) yielded average scores of 14.4 (SD = 6.74), 11.02 (SD = 4.85), and 16.6 (SD = 8.8), respectively [29–32]. It is important to note that none of the study samples was nationally representative, and hence it is not appropriate to make comparisons across these countries. With these caveats in mind, our study also found a high level of vaccine hesitancy among our surveyed population of social media users and calls for further investigation. While COVID-19 vaccine hesitancy has not been studied systematically using standardized measurement tools in African countries, limited evidence suggests a higher prevalence in Africa compared to other low- and middle-income settings [33, 34]. In addition, a plethora of studies across the globe have shown that the use of social media is strongly associated with increased COVID-19 vaccine hesitancy [35–37]. Our findings are in line with previous research and highlight the challenges of improving COVID-19 vaccine uptake on the African continent against this background [5–7].

To the best of our knowledge, this is one of the first evaluation studies of a social media-based communication campaign targeting COVID-19 vaccine-related attitudes and behaviors in African settings where we were able to demonstrate the potential of such a communication campaign to trigger positive changes in these domains. The major strengths of this study lie in

the robust empirical approach employed to quantify the campaign-attributable changes in the outcome and the use of validated scales to measure the important constructs associated with COVID-19 vaccine-related behaviors. Only one other study conducted in Nigeria in 2022 also employed a quasi-experimental design and demonstrated that the COVID-19 vaccination rate increased in the treatment population compared to the control population, corroborating our findings on the promising role of social media campaigns as an approach to increase COVID-19 vaccination rates in African settings. The overwhelming majority of the studies published during and after the COVID-19 pandemic, investigating the potential role of social media in changing COVID-19 vaccine-related attitudes, have employed cross-sectional observational study designs, lacking causal inference [38–41]. Particularly, studies evaluating the effectiveness of social media campaigns in changing vaccine-related attitudes or vaccine uptake were predominantly conducted in high-income countries [39–42]. Another strength of our study is the independent measurements of two distinct vaccine-related attitudes—vaccine hesitancy and vaccine confidence. Prior studies often lacked a clear distinction between vaccine hesitancy and confidence and used these attitude constructs inconsistently and interchangeably without clear definition or empirical measurement [43]. Considering the crucial role of vaccine-related attitudes in predicting and influencing individual vaccination decisions, it is essential to differentiate between vaccine hesitancy and confidence in research. This differentiation, along with measurements of these attitude constructs and vaccination behavior, is key to enhancing our understanding of the theoretical pathway to an individual's vaccination decision [43]. The lessons derived from our study can inform future evaluation studies of social media-based public health interventions.

The evaluation study had a number of limitations. First, the evaluation was conceived post-campaign. This limited the choice of the evaluation design and methods that could be used, which may have biased the estimated effects of the campaign, highlighting the importance of developing an evaluation plan before the start of the campaign. For example, to observe a 1% increase in vaccine uptake and account for a Type-I error cut-off of 0.05 and Type-II error cut-off of 0.8, power calculations suggest that a sample size of at least 9,525 per group per survey round would be required. Similarly, detecting a point reduction in vaccine hesitancy attributable to the campaign would require at least 1,006 samples per group per survey round. While a total of 3,775 and 2,694 respondents were recruited at pre- and post-campaign surveys, respectively, the relatively small sample size of 600 respondents in the control group might have resulted in insufficient power to detect significant effects for the purpose of the study.

Further, our sample was also skewed towards males, younger individuals, and those who were highly educated and who primarily obtained COVID-19-related information through social media. This was primarily because of the convenience sampling approach adopted, which relied on advertisement through social media, TV, and radio, resulting in an oversampling of individuals who would likely to be exposed to the campaign. This sampling bias had a few implications for the analysis. First, it may have increased the confounding effects of the demographic and socioeconomic characteristics on the campaign's impact. To account for this, we conducted our analyses using three different approaches—namely, crude, adjusted, and weighted-and-adjusted, and compared the results across these methods. In addition, by excluding important subpopulations, including those who did not have access to social media, older populations, females, and those whose educational attainment was less than secondary school, it is likely that the effect estimate was biased towards the null. Further, our data showed that the baseline proportion of respondents who self-reported to be fully vaccinated or partially vaccinated with a confirmed appointment(s) was considerably higher than the reported national vaccination coverage rate of 7.3% in June 2022 [14]. This discrepancy in the observed

and reported statistics may be partly attributed to our skewed samples, which disproportionately included individuals with higher socioeconomic status and those residing in the capital city of Dar es Salaam. Self-reported vaccination status might have also introduced an acquiescence bias and led to an over-reporting of the vaccination uptake in the study population [44]. Moreover, even if the campaign targeted a population of social media users in the country, it is important that the data for the evaluation come from an imbalanced sample across the treatment and control groups. This imbalance in sample characteristics also made the findings of the evaluation not generalizable to populations beyond our sample. The best way to ensure external validity in evaluation research is to use a representative sample of the population.

Another limitation of the evaluation stems from how we defined exposure to the campaign. Given the nature of the campaign, the collected programmatic data, and the fact that the evaluation study was conceived after the campaign was completed, it was not possible to clearly divide the surveyed population into an exposed and a non-exposed group for the purpose of a quasi-experimental evaluation study. Therefore, we used the information on respondents' main sources of COVID-19 information as a proxy to divide the respondents into a treatment and a control group. However, this approach may not accurately reflect actual exposure to the campaign and may have included in the treatment group the respondents who used social media as their main source of COVID-19 information but were not exposed to the campaign or who did not use social media as their main source of COVID-19 information but were still exposed to the campaign. The fact that the campaign took place on social media platforms and given its national scale and high intensity of exposure evidenced by the number of posts and mentions, the risk of reduced specificity in this classification approach is likely to be low. However, one cannot rule out the spillover effect of the campaign's messages to those who did not use social media, which could have resulted in reduced sensitivity. Therefore, the reduced accuracy in classification could have biased the DiD coefficient towards the null and may have resulted in an underestimation of the actual impact of the campaign.

In summary, the social-media based COVID-19 campaign in Tanzania demonstrated positive impact on vaccine hesitancy, vaccine confidence, and vaccine uptake among individuals aged 25–34 years, while also significantly reducing vaccine hesitancy among older adults aged 35 years and above. The differential impact of the campaign in improving COVID-19 vaccine confidence, hesitancy, and uptake across different age groups has implications for future implementation. First, our findings indicate the need for targeted campaigns that address the unique concerns of different age groups about COVID-19. Second, the success of the social media-based campaign attests to the potential of communication campaigns with positive messages about COVID-19 vaccines in improving attitudes toward COVID-19 vaccination [45]. Third, future studies should explore if social media-based campaigns combined with interventions targeting the key constructs of health behavior models, such as self-efficacy or perceived barriers, can lead to a change in vaccine uptake more effectively. Lastly, our study showed that some of the obstacles to conducting a rigorous evaluation can be mitigated by making the evaluation an integral part of the program planning process, highlighting the need for strong evaluation designs to produce reliable evidence for future implementation. In this regard, despite the aforementioned limitations, our study provides important insights for program implementers and decisionmakers considering to implement similar communication campaigns using social media platforms in the future.

## Supporting information

**S1 File.**
(DOCX)

**S2 File.**
(PDF)

**S3 File.**
(DOCX)

**S4 File.**
(DOCX)

**S5 File.**
(DOCX)

**S1 Checklist.** *PLOS ONE* **clinical studies checklist.**
(DOCX)

## Author Contributions

**Conceptualization:** Asad Lilani, Kate Campana.

**Data curation:** Sooyoung Kim, Asad Lilani, Caesar Redemptus, Kate Campana.

**Formal analysis:** Sooyoung Kim.

**Investigation:** Sooyoung Kim, Kate Campana.

**Methodology:** Sooyoung Kim.

**Project administration:** Asad Lilani, Caesar Redemptus, Kate Campana, Yesim Tozan.

**Resources:** Asad Lilani, Kate Campana, Yesim Tozan.

**Supervision:** Kate Campana, Yesim Tozan.

**Validation:** Sooyoung Kim.

**Visualization:** Sooyoung Kim.

**Writing – original draft:** Sooyoung Kim.

**Writing – review & editing:** Sooyoung Kim, Asad Lilani, Caesar Redemptus, Kate Campana, Yesim Tozan.

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
