## [Decision Letter · Decision Letter 0]

18 Dec 2023

PONE-D-23-21202A pre-post evaluation study of a social media-based COVID-19 communication campaign to improve attitudes and behaviors toward COVID-19 vaccination in TanzaniaPLOS ONE

Dear Dr. Tozan,

Thank you for submitting your manuscript to PLOS ONE. After careful consideration, we feel that it has merit but does not fully meet PLOS ONE’s publication criteria as it currently stands. Therefore, we invite you to submit a revised version of the manuscript that addresses the points raised during the review process.

We look forward to receiving your revised manuscript.

Kind regards,

Khin Thet Wai, MBBS, MPH, MA

Academic Editor

PLOS ONE

Journal Requirements:

Additional Editor Comments:

- To check and correct grammatical errors throughout the manuscript.

- To revise the methods, results, and discussion sections as required to improve scientific integrity.

Reviewers' comments:

Reviewer's Responses to Questions

**Comments to the Author**

1. Is the manuscript technically sound, and do the data support the conclusions?

Reviewer #1: Yes

Reviewer #2: Yes

2. Has the statistical analysis been performed appropriately and rigorously? 

Reviewer #1: Yes

Reviewer #2: Yes

3. Have the authors made all data underlying the findings in their manuscript fully available?

Reviewer #1: No

Reviewer #2: No

4. Is the manuscript presented in an intelligible fashion and written in standard English?

Reviewer #1: Yes

Reviewer #2: Yes

5. Review Comments to the Author

Reviewer #1: The authors are commended for their efforts in evaluating public health social media campaigns, given the complexity and novelty of the task. However, there are certain areas where the manuscript can be improved.

Methods:

1. To facilitate replication, the manuscript should provide more comprehensive information on the intervention's content and development. Including pictures or a link to videos (if applicable) would be beneficial.

2. From the outset, it should be explicitly stated that the populations in the pre- and post-evaluation phases are different.

3. Measurements:

- Elaborate on the scales used to measure hesitancy and confidence and their grading system to aid readers' understanding on what they evaluate and how they are different.

- Clearly define what was considered "fully vaccinated" during the study period, as this definition can vary across regions and time.

- Explain how vaccination status was measured and whether it relied on self-reporting.

- Provide further details on the rationale behind choosing the definition of exposure to the intervention in the study.

Results:

1. Clarify the equivalent of a "diploma” in school years since this term may not be universally understood.

2. Address the discrepancy between the study population size (pre and post) and the figure cited in the background (7.3%). Discuss potential reasons for this difference in the discussion section.

3. Reevaluate the presentation of vaccine confidence and hesitancy results to ensure clarity. If both measures increased after the intervention, clarify their underlying differences, and avoid discussing them in the same sentence.

4. Consider whether presenting Tables 2, 3, and 4 as written results without tables and replacing them with corresponding plots would enhance the manuscript's flow and readability.

5. Review the references to supplemental materials to ensure their accurate placement in the text since they also appear in the main body.

6. Correct the redundancy in line 268, "significantly significant."

Discussion:

1. Expand the discussion on vaccination-related topics, including the limitations of self-reporting and how it might differ from actual vaccination rates in the general population.

2. Include information on other approaches used to evaluate similar campaigns, providing readers with a broader perspective on the topic.

Reviewer #2: Ref: PONE-D-23-21202

A pre-post evaluation study of a social media-based COVID-19 communication campaign to improve attitudes and behaviors toward COVID-19 vaccination in Tanzania

Thank you for giving me the opportunity to review this interesting manuscript. Congratulations to the authors for their diligent work. While the manuscript is interesting, I have some queries as outlined below.

General comments

- The manuscript is well written in general. I appreciate that the authors prepared the manuscript according to the STROBE reporting checklist.

- The Methods section requires more information to ensure replicability.

- The Discussion section needs to be improved and deeper international discussion of the subject is required.

- Description of ethics statement is required as the study involved human participants. Any ethics approval from the IRB/ERC/REC? Was informed consent taken electronically or written?

Abstract

- Add objective in the background of the abstract.

- Sample size 5804 differs with (3443+2362 = 5805) throughout the manuscript. Is that typo? Please check.

Introduction

- The Introduction section is well written, and the study gap is clearly described.

Methods

- Is there any official hashtag, such as #AfricaCOVIDChampions mentioned in the introduction, that has been employed in Tanzania for the COVID-19 communication campaign? If so, please describe it.

- How did you approach the respondents? A detailed description of the data collection procedure should be presented. Any inclusion and exclusion criteria? Whether the data collection was conducted through social media channels? What specific tools were utilized—such as chat bots or Google Forms?

- Why pre campaign respondents exposed to treatment category?

- More variable explanation is required. For example, the operation definition of fully vaccinated in this study.

- To mention as “Oxford COVID-19 Vaccine Hesitancy Scale” to be comparable to other studies.

- Regarding Vaccine Confidence Scale questions, did the authors adapt from an existing reference, and if so, provide the relevant citation? Or were they developed for this study?

- How did you select variables to be included in the model? Any model fitting?

- How did you handle missing data as one-third of the outcome data is missing?

- I appreciate the use of propensity score matching method in the analysis.

- Regarding analysis in Vaccine Hesitancy and Vaccine Confidence Scales, how did you handle scoring if the respondents answered ‘Don’t Know’?

Results

- I recommend maintaining consistent decimal usage for p-values throughout the manuscript.

- Figure 2 provides the same information as Tables 2-4 and suggests removing the figure.

- I suggest providing the frequencies and percentages for each item in the Vaccine Hesitancy and Vaccine Confidence scales as supporting information tables.

Discussion

- Discussion should be improved by following the format 1) Summary of the main findings 2) Compare with previous studies 3) Strength and weakness 4) Implications.

- While the authors have conscientiously highlighted various limitations of the study, it is equally important to incorporate a discussion on its notable strengths.

- While the authors have provided a discussion on vaccine hesitancy from line 353 to 371, I observed a very few discussions regarding vaccine confidence and uptake in the discussion section. I suggest a more in-depth international discussion encompassing all three outcomes.

6. PLOS authors have the option to publish the peer review history of their article (what does this mean?). If published, this will include your full peer review and any attached files.

Reviewer #1: **Yes: **Lucía Abascal Miguel

Reviewer #2: **Yes: **Kyaw Lwin Show

---

## [Author Response · Author response to Decision Letter 0]

6 Feb 2024

Response to reviewers: PONE-D-23-21202

Reviewer 1

Reviewer #1: The authors are commended for their efforts in evaluating public health social media campaigns, given the complexity and novelty of the task. However, there are certain areas where the manuscript can be improved.

Methods:

1. To facilitate replication, the manuscript should provide more comprehensive information on the intervention's content and development. Including pictures or a link to videos (if applicable) would be beneficial.

-> Thank you for your feedback. For readers who are interested in the intervention’s content, we have included a link to the campaign's website where the toolkits are publicly available for download (lines 129-131). Additionally, we have added a Supporting Information section that presents detailed information on the media coverage of the campaign, which includes photos and videos (lines 131-132).

2. From the outset, it should be explicitly stated that the populations in the pre- and post-evaluation phases are different.

-> Thank you for this suggestion. We have clarified this in the beginning of the Methods section where we described the data collection approach (lines 136-138).

3. Measurements:

- Elaborate on the scales used to measure hesitancy and confidence and their grading system to aid readers' understanding on what they evaluate and how they are different.

-> Thank you for this comment. The measurement scales are explained in detail in the Methods section (lines 144-158) and the full instruments are provided in the Supporting Information 2 (lines 157-158). We have further clarified that we used the Oxford COVID-19 vaccine hesitancy and confidence scales and provided a citation that details the development and content of these two scales (lines 146-150)

- Clearly define what was considered "fully vaccinated" during the study period, as this definition can vary across regions and time.

-> Thank you for this suggestion. This has now been clarified in lines 140-142.

- Explain how vaccination status was measured and whether it relied on self-reporting.

-> Thank you for this suggestion. We have now clarified that our data on vaccination status relied on self-report (line 140).

- Provide further details on the rationale behind choosing the definition of exposure to the intervention in the study.

-> Thank you for this suggestion. We have included additional explanation on how we defined exposure to the intervention in the Methods section (lines 176-180) and discussed the resulting limitations in the Discussion section (lines 391-406).

Results:

1. Clarify the equivalent of a "diploma” in school years since this term may not be universally understood.

-> Thank you for this suggestion. We clarified in lines 216-218 that diploma in the Tanzanian school system is equivalent to the Advanced Certificate of Secondary Education (ACSE) or the grades 13-14. The same information is also added as a footnote in Table 1.

2. Address the discrepancy between the study population size (pre and post) and the figure cited in the background (7.3%). Discuss potential reasons for this difference in the discussion section.

-> Thank you for this suggestion. We believe that the discussion section would be a more appropriate place to address this comment, and we have provided further explanation about this discrepancy in this section (lines 379-385).

3. Reevaluate the presentation of vaccine confidence and hesitancy results to ensure clarity. If both measures increased after the intervention, clarify their underlying differences, and avoid discussing them in the same sentence.

-> Thank you for this suggestion. The overall results reporting the campaign-attributable change in vaccine confidence and hesitancy are already presented separately. We have revised the paragraph summarizing the age-stratified results to clearly present the changes in vaccine confidence, hesitancy, and uptake (lines 270-288).

4. Consider whether presenting Tables 2, 3, and 4 as written results without tables and replacing them with corresponding plots would enhance the manuscript's flow and readability.

-> Thank you for this comment. The main findings from Tables 2 to 4 are summarized in the manuscript in lines 250-264. In response to the reviewer's suggestion that Tables 2 to 4 and Figure 2 might be redundant, we have opted to present the tables in the Supporting Information.

5. Review the references to supplemental materials to ensure their accurate placement in the text since they also appear in the main body.

-> Thank you for this suggestion. We have reviewed the manuscript to ensure that all the supplementary materials are correctly cited throughout the text.

6. Correct the redundancy in line 268, "significantly significant."

-> Thank you for catching this. It is now corrected.

Discussion:

1. Expand the discussion on vaccination-related topics, including the limitations of self-reporting and how it might differ from actual vaccination rates in the general population.

-> Thank you for these valuable suggestions. We have significantly revised our discussion section to improve on the specific point you raised. As a result, the expanded discussion on vaccination-related topics is now covered in lines 387-394. 

2. Include information on other approaches used to evaluate similar campaigns, providing readers with a broader perspective on the topic.

-> Thank you for this suggestion. We have significantly revised the discussion section to address the comments from all reviewers. A comparison of our approach with the approaches used in other published studies aimed at evaluating similar campaigns is now provided in lines 340-364.

Reviewer 2

A pre-post evaluation study of a social media-based COVID-19 communication campaign to improve attitudes and behaviors toward COVID-19 vaccination in Tanzania

Thank you for giving me the opportunity to review this interesting manuscript. Congratulations to the authors for their diligent work. While the manuscript is interesting, I have some queries as outlined below.

General comments

- The manuscript is well written in general. I appreciate that the authors prepared the manuscript according to the STROBE reporting checklist.

- The Methods section requires more information to ensure replicability.

- The Discussion section needs to be improved and deeper international discussion of the subject is required.

- Description of ethics statement is required as the study involved human participants. Any ethics approval from the IRB/ERC/REC? Was informed consent taken electronically or written?

-> Thank you. We hope that the revisions we made have successfully addressed all of the above points. Please see our response below for the specific revisions we made based on your comments.

Abstract

Add objective in the background of the abstract.

-> Thank you. We have added the study objective into the abstract (Lines 30-32).

Sample size 5804 differs with (3443+2362 = 5805) throughout the manuscript. Is that typo? Please check. 

-> Thank you for catching this. We corrected the typo made in the pre-campaign sample size to the correct figure (3,442).

Introduction

The Introduction section is well written, and the study gap is clearly described.

-> Thank you for this encouraging comment.

Methods

Is there any official hashtag, such as #AfricaCOVIDChampions mentioned in the introduction, that has been employed in Tanzania for the COVID-19 communication campaign? If so, please describe it. 

-> Thank you for this suggestion. We have included a link to the campaign's website in the manuscript (lines 129-131), where toolkits are publicly available for download. Additionally, we have added a Supporting Information section presenting the media and social media coverage of the campaign, including photos and videos (lines 131-132). The official hashtags used for the campaign are also provided in the Supporting Information. 

How did you approach the respondents? A detailed description of the data collection procedure should be presented. Any inclusion and exclusion criteria? Whether the data collection was conducted through social media channels? What specific tools were utilized—such as chat bots or Google Forms?

-> Thank you for this suggestion. We used a convenience sampling approach via an online survey platform, and we have revised the paragraph explaining the sampling approach and the inclusion criteria to provide further details (lines 159-167). The analysis was carried out using R software, and this information is included in lines 201-202.

Why pre campaign respondents exposed to treatment category?

-> In a quasi-experimental difference-in-difference design, it is necessary to assign all respondents into a treatment and a control group prior to their exposure to the intervention. That way, it becomes possible to measure and compare the changes in outcomes over time between the treatment and control groups, allowing for a more robust analysis of the intervention's impact by accounting for pre-existing differences between the groups. At the time of conceiving this evaluation study, programmatic data collection at national level was already completed, and there was no clearly defined counterfactual group (i.e., a population not exposed to the campaign). Therefore, we used the information on respondents’ main sources of COVID-19 information as a proxy to divide the respondents into a treatment and a control group at baseline and endline to be able to employ a quasi-experimental DiD analysis. We have included additional explanation on how we defined exposure to the intervention in the Methods section (lines 176-180) and discussed the resulting limitations in the Discussion section (lines 391-406). 

More variable explanation is required. For example, the operation definition of fully vaccinated in this study.

-> Thank you for this suggestion. We have clarified the definition of “fully vaccinated” in the manuscript (lines 140-142)

To mention as “Oxford COVID-19 Vaccine Hesitancy Scale” to be comparable to other studies.

-> This is now clarified in the manuscript (lines 146-150).

Regarding Vaccine Confidence Scale questions, did the authors adapt from an existing reference, and if so, provide the relevant citation? Or were they developed for this study?

-> We used the Oxford COVID-19 Vaccine Confidence and Complacency Scale without any modification. This is now clarified in the manuscript with a citation to the relevant publication (lines 146-150).

How did you select variables to be included in the model? Any model fitting?

-> Thank you for this question. According to the existing literature, socioeconomic factors may be significant confounders of COVID-19 vaccine hesitancy, confidence, and uptake. Therefore, we integrated the socioeconomic variables identified as potential confounders—indicated by a significant p-value in the descriptive analysis—into our model. We have now clarified this point in the manuscript (lines 191-196). 

How did you handle missing data as one-third of the outcome data is missing?

-> Thank you for your question. In the regression models, only observations with complete data for all the variables included in the model were utilized. The limitations due to the convenience sampling method and the small analytical sample size, which may result in insufficient power and bias toward the null, are extensively discussed in the Discussion section (lines 368-390).

I appreciate the use of propensity score matching method in the analysis.

-> Thank you. We included the results from the matched analysis to ensure our findings are robust.

Regarding analysis in Vaccine Hesitancy and Vaccine Confidence Scales, how did you handle scoring if the respondents answered ‘Don’t Know’?

-> Thank you for this question. We followed the instructions of the authors who developed the scale, and excluded the “Don’t Know” answers from the analysis by assigning a score of 0. 

Results

I recommend maintaining consistent decimal usage for p-values throughout the manuscript. 

-> Thank you. We have reported all p-values up to the third decimal point across the manuscript. We also have updated the manuscript to report the correct coefficients and p-values from the corresponding tables.

Figure 2 provides the same information as Tables 2-4 and suggests removing the figure.

-> Thank you for this suggestion. We appreciate the feedback and have decided, in line with the reviewers' recommendations, to move Tables 2~4 to the Supporting Information. The key findings from these tables are now summarized in the manuscript (lines 250-264).

I suggest providing the frequencies and percentages for each item in the Vaccine Hesitancy and Vaccine Confidence scales as supporting information tables.

-> They are now provided in the Supporting Information 4 (Tables S4-5 and 6) and referenced in the main manuscript (lines 235-238).

Discussion

Discussion should be improved by following the format 1) Summary of the main findings 2) Compare with previous studies 3) Strength and weakness 4) Implications.

-> Thank you for this suggestion. We have followed your suggestion and have substantially revised the Discussion section. Now the summary of the main findings is stated in lines 303-322, followed by a comparison of our findings with previous studies and a summary of our study’s strengths (lines 323-364), limitations (lines 365-414), and implications (lines 415-431). 

While the authors have conscientiously highlighted various limitations of the study, it is equally important to incorporate a discussion on its notable strengths. 

-> Thank you for this suggestion. We revised the Discussion section to clearly articulate the strengths of our study (lines 323-364). 

While the authors have provided a discussion on vaccine hesitancy from line 353 to 371, I observed a very few discussions regarding vaccine confidence and uptake in the discussion section. I suggest a more in-depth international discussion encompassing all three outcomes.

-> Thank you for this valuable suggestion. We have observed that many existing studies on COVID-19 vaccine-related attitudes have often interchangeably used terminologies, such as vaccine hesitancy and confidence, with the majority of the literature preferring to use "vaccine hesitancy" as an overarching concept rather than a clearly defined construct. We believe that one of the strengths of our study is the clear differentiation of these two constructs and their measurement alongside the behavioral construct of vaccine uptake. Further elaboration on this point can be found in the revised Discussion section (lines 337-345) in view of your comments.

---

## [Decision Letter · Decision Letter 1]

14 Feb 2024

PONE-D-23-21202R1A pre-post evaluation study of a social media-based COVID-19 communication campaign to improve attitudes and behaviors toward COVID-19 vaccination in TanzaniaPLOS ONE

Dear Dr. Tozan,

Thank you for submitting your manuscript to PLOS ONE. After careful consideration, we feel that it has merit but does not fully meet PLOS ONE’s publication criteria as it currently stands. Therefore, we invite you to submit a revised version of the manuscript that addresses the points raised during the review process.

 Please submit your revised manuscript by Mar 30 2024 11:59PM. If you will need more time than this to complete your revisions, please reply to this message or contact the journal office at plosone@plos.org. Please include the following items when submitting your revised manuscript:A rebuttal letter that responds to each point raised by the academic editor and reviewer(s). You should upload this letter as a separate file labeled 'Response to Reviewers'.A marked-up copy of your manuscript that highlights changes made to the original version. You should upload this as a separate file labeled 'Revised Manuscript with Track Changes'.An unmarked version of your revised paper without tracked changes. You should upload this as a separate file labeled 'Manuscript'.If applicable, we recommend that you deposit your laboratory protocols in protocols.io to enhance the reproducibility of your results. Protocols.io assigns your protocol its own identifier (DOI) so that it can be cited independently in the future. For instructions see: https://journals.plos.org/plosone/s/submission-guidelines#loc-laboratory-protocols. Additionally, PLOS ONE offers an option for publishing peer-reviewed Lab Protocol articles, which describe protocols hosted on protocols.io. Read more information on sharing protocols at https://plos.org/protocols?utm_medium=editorial-email&utm_source=authorletters&utm_campaign=protocols.

We look forward to receiving your revised manuscript.

Kind regards,

Khin Thet Wai, MBBS, MPH, MA

Academic Editor

PLOS ONE

Journal Requirements:

**Additional Editor Comments:**

Please follow the comments of the reviewer.

Reviewers' comments:

Reviewer's Responses to Questions

**Comments to the Author**

1. If the authors have adequately addressed your comments raised in a previous round of review and you feel that this manuscript is now acceptable for publication, you may indicate that here to bypass the “Comments to the Author” section, enter your conflict of interest statement in the “Confidential to Editor” section, and submit your "Accept" recommendation.

Reviewer #2: (No Response)

2. Is the manuscript technically sound, and do the data support the conclusions?

Reviewer #2: Yes

3. Has the statistical analysis been performed appropriately and rigorously? 

Reviewer #2: Yes

4. Have the authors made all data underlying the findings in their manuscript fully available?

Reviewer #2: Yes

5. Is the manuscript presented in an intelligible fashion and written in standard English?

Reviewer #2: Yes

6. Review Comments to the Author

Reviewer #2: - Description of ethics statement is mandatory as the study involved human participants. Any ethics

approval from the IRB/ERC/REC? Was informed consent taken electronically or written? etc...

- LINE 192-196 It is unclear whether you selected the variables to be included in the adjusted model based on previous literatures or used p-value from the crude analysis.

7. PLOS authors have the option to publish the peer review history of their article (what does this mean?). If published, this will include your full peer review and any attached files.

Reviewer #2: No

---

## [Author Response · Author response to Decision Letter 1]

16 Feb 2024

Thanks very much for this round of comments. Please see our point-by-point responses below. 

Reviewer #2: - Description of ethics statement is mandatory as the study involved human participants. Any ethics approval from the IRB/ERC/REC? Was informed consent taken electronically or written? etc...

As stated in lines 107 – 109, “This study was a secondary analysis of the pre-and post-campaign data collected by the Access Challenge (TAC), the implementing agency of the campaign, for their internal monitoring and evaluation purposes.” We received de-identified data from the implementing agency for the purpose of this research, and an Institutional Review Board (IRB) review was deemed not required by New York University’s IRB. We further clarified this information in lines 135-137 and provided an NYU internal form confirming this information as Supporting Information 2. Please also note that “Informed consent was sought from all respondents using an electronic survey form at the time of the original data collection by the implementing agency, and only consented individuals participated in the survey.” This is clarified in lines 170-172.

LINE 192-196 It is unclear whether you selected the variables to be included in the adjusted model based on previous literatures or used p-value from the crude analysis.

As stated in the lines 192-196, we used both literature review and crude statistical tests during the descriptive analysis to select the variables.

---

## [Editor Report · Decision Letter 2]

23 Feb 2024

A pre-post evaluation study of a social media-based COVID-19 communication campaign to improve attitudes and behaviors toward COVID-19 vaccination in Tanzania

PONE-D-23-21202R2

Dear Dr. Tozan,

We’re pleased to inform you that your manuscript has been judged scientifically suitable for publication and will be formally accepted for publication once it meets all outstanding technical requirements.

Kind regards,

Khin Thet Wai, MBBS, MPH, MA

Academic Editor

PLOS ONE

Additional Editor Comments (optional):

All comments are adequately addressed.